# What is the feasibility and patient acceptability of a digital system for arm and hand rehabilitation after stroke? A mixed-methods, single-arm feasibility study of the 'OnTrack' intervention for hospital and home use

Gianpaolo Fusari ,[1] Ella Gibbs,[1] Lily Hoskin,[1] Anna Lawrence-Jones,[1] Daniel Dickens,[1] Roberto Fernandez Crespo ,[2] Melanie Leis,[2] Jennifer Crow,[3] Elizabeth Taylor ,[4] Fiona Jones,[4] Ara Darzi[5]

For numbered affiliations see end of article.

**Correspondence to**
Gianpaolo Fusari;
gianpaolo@helixcentre.com

## ABSTRACT

**Objectives** Arm weakness is common after stroke; repetitive activity is critical for recovery but people struggle with knowing what to do, volume, and monitoring progress. We studied the feasibility and acceptability of OnTrack, a digital intervention supporting arm and hand rehabilitation in acute and home settings.

**Design** A mixed-method, single-arm study evaluating the feasibility of OnTrack for hospital and home use. An independent process evaluation assessed the intervention's fidelity, dose and reach. Amendments to the protocol were necessary after COVID-19.

**Setting** Acute stroke services and home settings in North West London.

**Participants** 12 adults with a stroke diagnosis <6 months previously (first or recurrent) requiring arm rehabilitation in hospital and/or home.

**Intervention** 12 weeks using the OnTrack system comprising arm tracking and coaching support for self-management.

**Primary and secondary outcome measures** Recruitment, retention and completion rates; compliance and adherence to the intervention; reasons for study decline/withdrawal.

Intervention fidelity and acceptability, evaluated through an independent process evaluation.

Patient measures including activity baseline, healthcare activation, arm function and impairment collected at baseline, week 7 and week 14 of participation to assess suitability for a randomised controlled trial (RCT).

**Results** 181 individuals screened, 37 met eligibility criteria, 24 recruited (65%); of these, 15 (63%) were recruited before COVID-19, and 9 (37%) during. 12 completed the intervention (50%). Despite COVID-19 disruptions, recruitment, retention and completion were in line with prestudy expectations and acceptable for a definitive trial. Participants felt the study requirements were acceptable and the intervention usable. Fidelity of delivery was acceptable according to predetermined fidelity markers. Outcome measures collected helped determine sample size estimates and primary outcomes for an RCT.

**Conclusions** The intervention was found to be usable and acceptable by participants; study feasibility objectives were met and demonstrated that a definitive RCT would be viable and acceptable.

**Trial registration number** NCT03944486.

### STRENGTHS AND LIMITATIONS OF THIS STUDY

⇒ This was the first feasibility trial of a novel intervention employing an integrated approach for tracking arm activity and self-management coaching with the aim of increasing the opportunities for independent rehabilitation.

⇒ Recruitment for the study began in September 2019, modifications to the protocol were necessary to enable the delivery of the intervention remotely after the start of the COVID-19 19 pandemic in March 2020.

⇒ A patient and public involvement group met four times during the study; the group gave advice regarding modifications to the protocol, and contributed to the interpretation and dissemination of data findings.

⇒ An independent process evaluation was carried out to provide detailed information about implementation, context, and the mechanisms of impact of the intervention.

⇒ Longer-term follow-up of participants was not possible within the time frame set for the trial; however, participant views were sought regarding the acceptability of longer-term follow-up.

## INTRODUCTION
### Background

Globally, five million people are left disabled from stroke, commonly with a form of arm

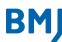

impairment.[1] In the UK, 75% of disabled stroke survivors, or ~4 50 000 people, have an arm weakness.[2] By 2035, this number may increase by a third.[3] Stroke costs the UK society £26b annually, by 2025 costs could increase to £43b and to £75b by 2035. An ageing population, better stroke survival rates, and the overall increase in labour costs account for this trend.[3]

Understanding when to provide interventions that can improve arm function is recognised as a national research priority[4]; despite this, time spent providing therapy for the arm is often limited, resulting in patients spending minimal time rehabilitating or being active.[5 6] There is a correlation between physical activity and the ability to perform activities of daily living using the arm,[7] but a Cochrane review of over 500 trials failed to yield high-quality practice recommendations for arm interventions.[8] The COVID-19 pandemic exacerbated this problem resulting in thousands of stroke survivors receiving diminished rehabilitation and an increase in health inequalities.[9 10] Without specialist support, rehabilitation can be less effective and an isolating experience.[11]

Ethnographic studies describe how patients struggle to see and keep track of improvements[12–14]—especially outside of scheduled one-to-one therapy—having an impact on motivation and creating dependency on therapists for feedback.[6 7 15 16] Stroke survivors often report feeling unsupported after leaving hospital and not knowing how to best help themselves improve their arm function.

We believe repetitive activity can be increased by targeting the time patients go about their daily activities and could use their arm movement (however, small) to a greater extent. Capacity for activity could be increased further by using self-management strategies as demonstrated by several programmes in stroke and other long-term conditions.[17–20] In addition, there is strong evidence to suggest that digital interventions,[21] and in particular digital therapeutics (DTx),[22] could help support stroke patients in their rehabilitation and life after stroke.

The aim of this study was to assess the feasibility of the OnTrack intervention—which could increase opportunities for arm activity by improving individuals' self-management skills through tailored support and real-time activity feedback—and inform the design of a definitive randomised controlled trial (RCT).

As per study protocol,[23] recruitment, adherence and retention rates were measured, as well as reasons for declining participation or withdrawing from the trial; the utility and completion of outcome measures to provide an indication of effectiveness to inform sample size calculations for an RCT; the usability of OnTrack by study participants; and the acceptability of study procedures by front-line staff. An independent process evaluation assessed the fidelity of the intervention delivery as well as the acceptability of the intervention to participants and delivery team.

## METHODS

### Design

A feasibility study with a nested process evaluation. The study was a single-site, non-randomised intervention trial. The design of the study was developed through a collaborative approach between the authors, a patient and public involvement (PPI) steering group, stroke therapists (occupational therapists and physiotherapists) and the Research Design Service at the National Institute for Health Research.

### Study settings

Prior to lockdown restrictions caused by the COVID-19 pandemic, participants were enrolled in the study at an inner city National Health Service (NHS) hospital Trust in West London and, where necessary, continued their participation at home after discharge to complete the 14-week study period. After restrictions were introduced, all participants were enrolled at home and were followed remotely for 14 weeks.

### Participants

Participant inclusion and exclusion criteria are fully defined in the study protocol[23]; briefly, participants were adults with a stroke diagnosis less than 6 months previously (first or recurrent) requiring arm and hand rehabilitation in hospital and/or at home, medically stable, presenting without arm pain or oedema, and with capacity to consent.

### Recruitment

Participants were recruited from the Hyperacute Stroke Unit, Acute Stroke Unit and Clinical Neuro Rehabilitation Unit at an NHS Hospital Trust in London.

Recruitment followed two distinct procedures before and after lockdown restrictions were in place. Between August 2019 and March 2020, hospital therapy teams were responsible for screening, introducing the study and providing information documents to potential participants. After taking consent, therapists shared patient information with the research team. Recruitment was suspended between March and August 2020. From September 2020 onwards, therapy teams continued to perform eligibility screens sharing patient information with the research team only at discharge. The consent and recruitment process was then completed remotely at participants' homes.

### Intervention

Participants were enrolled in the study for 14 weeks. Week 1 consisted of baseline assessments; weeks 2–13 consisted of the OnTrack intervention (detailed below); week 14 consisted of follow-up assessments and an interview performed by the process evaluation team. Additionally, a mid-participation assessment was performed during week 8.

The intervention was the OnTrack system, consisting of three main components:

**Table 1** In-app messages

| Messages and links to content related to stroke rehabilitation and self-management were sent to participants via the OnTrack smartphone application. Messages were divided into four different categories. | | | |
|---|---|---|---|
| Message category | Description | Frequency, time of day | Sample message |
| Intention settings | Message to set the participant's intentions for the day. Participants were able to respond to these messages with a yes/no answer. | Weekdays, in the morning | Good morning Jon! Aim to wear the watch on your LEFT wrist for as long as you can today and keep an eye on your arm activity target of 45 minutes. Are you ready to try this? |
| Tips and advice | Personalised messages to include tips or advice relevant to the participant's situation. This type of message did not require a response. | 2–3 times per week, at midday | Weekends are often different from the rest of the week but you can still include activities that involve your LEFT arm. It's also ok to have a rest but don't forget to use appropriate cutlery for every meal if you have the opportunity as we discussed before. You've got this! |
| Reflective practice | Message with the intention to help participants reflect on their progress. This type of message did not require a response. | 2–3 times per week, towards the end of the day | Hi Jon! I hope you managed to reach your target of 45 minutes of arm activity today. Were you able to think about new ways of involving your arm? Let's talk about these when we meet! |
| Links to external content | Participants in phase 1 of the study received a total of 9 links while participants in phase 2 received 11 links signposting to resources they could tap into once their participation ended. | 9–11 links sent over 14 days, at midday | Hi Jon, the Stroke Foundation in Australia have put together a great blog post on how to improve your Problem Solving skills. Click here to check it out when you get a chance! (link) |

1. Activity tracking. All participants were asked to wear a smartwatch on their affected arm during waking hours (typically 12 hours) which included software and a purpose-built algorithm providing data on gross arm activity (minutes of activity performed).
2. Motivational content. Delivered as visual feedback on the amount of activity performed, personalised in-app messages, and links to educational material. A sample of content can be seen in table 1.
3. Self-management coaching. Delivered during 1–1 coaching sessions (in person and remotely). The coaching component has been influenced by the self-management principles as defined by the Bridges Self-Management Programme[24] and the Taking Charge After Stroke[25] self-management programme. Details of the coaching component are provided as online supplemental file.

Activity data collected by participants was monitored remotely by the research team, this helped to guide conversations during coaching sessions.

The coaching component was reviewed at the study's halfway point following data analysis performed by the process evaluation team in discussion with the PPI group.

Participants were loaned all equipment necessary for the trial and no previous experience with using smart devices was required to participate. Technical support was provided by the intervention team in cases where the hardware and/or software failed to perform the required functions to deliver the intervention.

## Outcomes

### Feasibility of trial design and procedures

Outcomes included the measurement of recruitment rates, including number of patients screened, eligible, consented and excluded after screening; participant adherence to the intervention and usage (percentage of days using OnTrack, minutes of activity as recorded using OnTrack, engagement with features in the OnTrack app); completion rates; and acceptability and reasons for decline/withdrawal.

NHS therapists responsible for screening and recruitment were invited to complete a survey to gather their feedback regarding acceptability of study procedures.

### Clinical assessments

Clinical outcomes were collected to identify an appropriate primary outcome, and to estimate parameters for a sample size calculation for an RCT, these were collected at the start, halfway and end of participation (table 2).

### Additional assessments

Activity baselines were gathered at weeks 1 and 14 of participation, study participants wore activity trackers (Axivity AX3) on both arms during waking hours (typically 12 hours/day) for 3–7 days to gather activity baseline data and allowing for left-right arm usage comparison.

The System Usability Scale[26] was used to subjectively assess the usability of the OnTrack intervention.

**Table 2** Outcome measure schedule

| Concept | Assessment | Week performed |
| --- | --- | --- |
| Patient activation/ engagement | Patient Activation Measure[33 34] | 1, 8*, 14 |
| Arm impairment | Fugl-Meyer Assessment for upper extremity (FMA-UE)†[35 36] | 1, 8*, 14 |
| Arm function | Upper-Extremity Motor Activity Log-14[30] | 1, 8*, 14 |
| Gross level of disability | Modified Rankin Scale[37] | 1, 8*, 14 |
| Arm pain | Visual Analogue Scale (VAS)[38] | 1, 8*, 14 |
| Cognitive impairment | Montreal Cognitive Assessment‡[39 40] | 1, 8*, 14 |
| Arm neglect | Albert's Test (AT)[41] | 1, 8*, 14 |
| Quality of life | EQ-5D-5L[42] | 1, 8*, 14 |
| Activity baseline | Axivity AX3 usage on both arms | 1, 14 |
| System usability | System Usability Scale[26] | 14 |

*Performed at week 7 during phase 2 of the study.
†Not performed during phase 2 due to inability to administer remotely.
‡A modified version was performed during phase 2.
EQ-5D-5L, EuroQol-5 Dimensions-5 Level.

## Process evaluation

An independent process evaluation was conducted in parallel to this trial. The evaluation aimed to determine whether the intervention was delivered as intended and to understand the mechanisms of impact. Interviews were conducted with participants finishing the intervention period. These focused on their experience of using OnTrack and the different components of the intervention, as well as on their perceptions of the impact OnTrack had on their rehabilitation and ability to self-manage.

## Sample size

The sample size was determined using guidelines that advocate a sample size between 12 and 30 for feasibility studies.[27] We aimed to recruit a minimum of 24 participants over the duration of the study which was in line with referral rates at the recruiting site.

## Analysis

Analysis was performed on the study parameters and its implementation. We evaluated the usage of OnTrack as well as outcome measures and recorded changes over time.

Patterns of activity were analysed by day and hour of day. Comparisons between activity data and usage of different application features were created to understand their influence in activity output. Similarly, usage of individual features were recorded (eg, engagement with messages, number of times activity stats were accessed).

Furthermore, a comparison between activity minutes and performance in the different outcome measures was performed.

Pearson correlation coefficient was used to explore relationships between application usage and patient activity. A multivariate logistic regression model was used to investigate the independent effects of the the daily activity time of each patient, measured in 30 min intervals ('Activity'), the number of days since the patient enrolment in the trial ('Time'), and whether the patient suffered a stroke on their dominant arm ('Dominant arm') on the patients' left-right arm usage ratio. Multivariate linear regression models were used to assess the independent effects of the 'Activity' and 'Time' variables on the other outcome measures in this study.

Data collected for the process evaluation was a combination of qualitative data from semistructured interviews and observations of coaching sessions. At the study half-point, a workshop was conducted with the PPI group who analysed interview transcripts and made recommendations to refine the protocol.

### Patient and public involvement

A PPI group comprising three stroke survivors helped refine the intervention to the version used for this study. The group met a total of four times over the duration of the study, their time and travel were reimbursed according to INVOLVE[28] guidelines. The group supervised the development of all patient-facing material to ensure its clarity and accessibility. Members were trained by experienced researchers to participate in qualitative data analysis at the study's halfway point. They helped to refine themes and key messages arising from qualitative interviews. The PPI involvement plan was shared with Imperial College London's PPI 'Research Partners Group' to ensure the needs of the steering group were accounted for.

## RESULTS
### Recruitment

Recruitment took place between August 2019 and December 2020 with an imposed study suspension between March 2020 and August 2020 due to the public health emergency caused by the COVID-19 pandemic.

In total, 181 individuals were screened for eligibility with 37 meeting the inclusion criteria invited to participate (figure 1). 24 participants were recruited with a response rate of 65%, of these, 15 (63%) were recruited before the study suspension and 9 (37%) between September and December 2020.

### Participant characteristics

Data collected from participants during the initial assessment was available for 21 participants (11 females; mean age 61.1, range 33.5–82.5), three participants were lost to follow-up before a session could be arranged. For the majority of participants (n=16, 76%) this was their first stroke, only one participant had had more than two

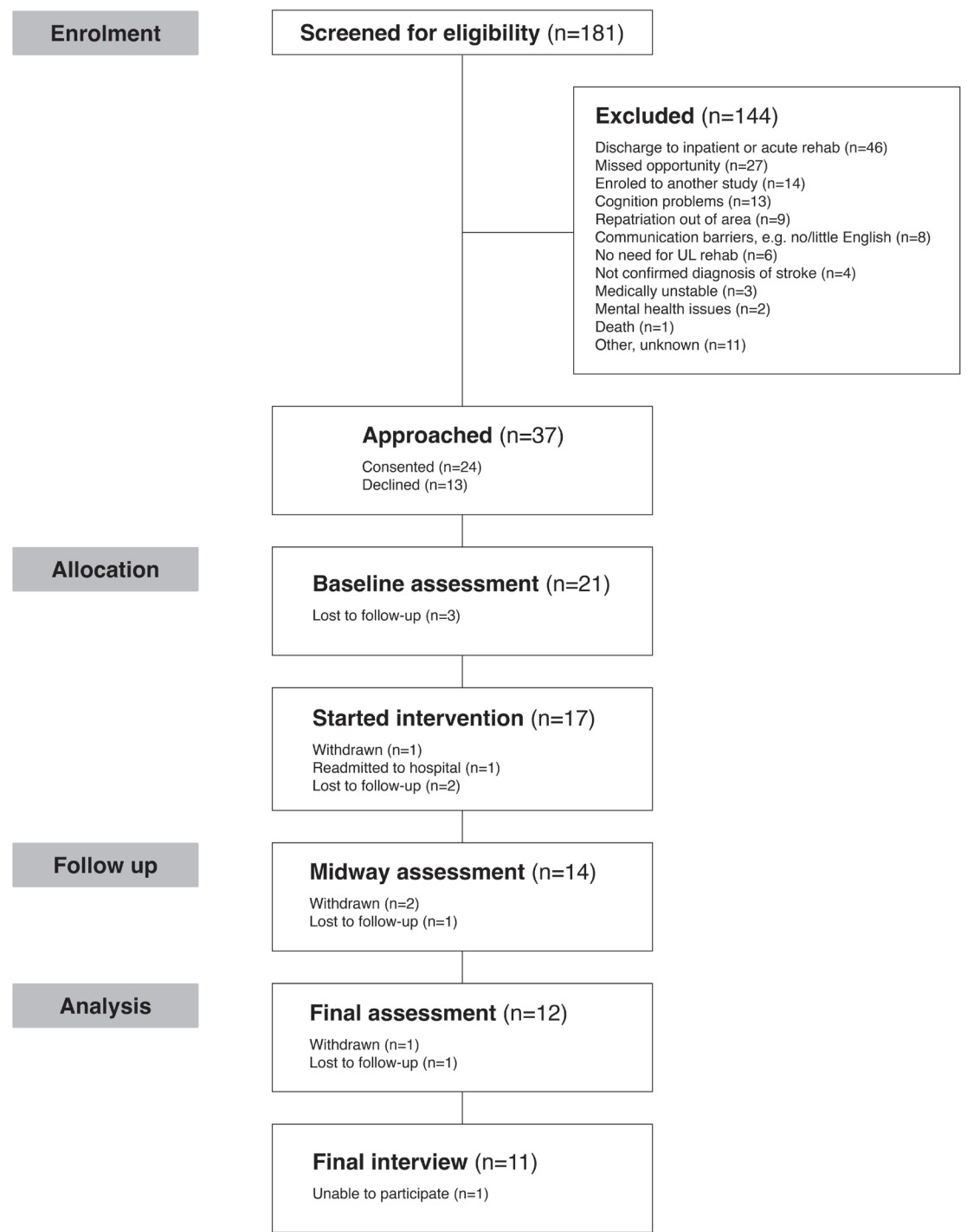

**Figure 1** CONSORT flow diagram. CONSORT, Consolidated Standards of Reporting Trials. UL = Upper Limb

previous strokes. Onset of the last stroke varied between 11 and 141 days with a mean of 42.2 days. Seven participants (33%) were impaired on their dominant arm. All participants owned or had experience with mobile phones but only 15 (71%) were smartphone users; 18 (86%) had WiFi at home. Although this was not a randomised study, participants' acceptance of randomisation was queried; 13 (62%) expressed they would have consented to the study regardless of whether they'd be allocated to a control or intervention group.

All participants underwent some form of rehabilitation treatment by a UK NHS acute and/or community provider, ranging from in-person Early Supported Discharge (ESD) to remote community neurorehabilitation. However, the study did not collect detailed information on the type of rehabilitation that was provided.

Participant baseline characteristics can be seen in table 3.

### Adherence and retention

Of the 24 participants recruited, 21 (87%) took part in an initial session with researchers to gather baseline

| Table 3 | Participant characteristics at baseline (n=21) |
|---|---|
| **Gender** | **11 female (52%)** |
| Age (years); mean (SD); median (min, max) | 61.1 (12.5); 60.5 (33.5 min, 82.5 max) |
| Ethnicity; n (%) | |
| White British | 7 (33) |
| White other | 3 (14) |
| Asian | 5 (25) |
| Black | 3 (14) |
| Other | 2 (10) |
| Prefer not to say | 1 (5) |
| Impaired arm | 7 Right (33) |
| Dominant arm | 19 Right (90) |
| Dominant arm impaired | 7 Yes (33) |
| Stroke onset (days); mean (SD); median (min, max) | 42.2 (33.04); 33 (11 min, 141 max) |
| Type of stroke; n (%) | |
| Ischaemic | 11 (52) |
| Haemorrhagic | 2 (10) |
| Unknown | 8 (38) |
| First stroke | 16 Yes (76) |
| Comorbidities | 15 Yes (71) |
| Smoker | 5 Yes (24) |
| Smartphone user | 15 Yes (71) |
| WiFi at home | 18 Yes (86) |
| Amenable to randomisation | 13 Yes (62), 2 No (10), 6 Don't know (29) |

characteristics; 18 (75%) completed outcome measures and 17 (71%) went through to complete the activity baseline week. Of the seven participants dropping out at this stage, researchers were unable to make contact with five of them; one was readmitted to hospital; and one decided to stop participation after feeling numbness on the arm.

Seventeen (71%) participants started using OnTrack; 14 (58%) repeated outcome measures at halfway (weeks 7–8 of participation) and 12 (50%) completed the intervention period, repeated outcome measures and completed the final activity baseline week. Of the five participants

lost between starting OnTrack and finishing the study, three withdrew consent and two were lost to follow-up. Reasons for withdrawing consent included one person who felt they had gained all they could from OnTrack and wanted to continue on their own; another felt that the intervention would not help them as they had too much on their mind; and another felt too anxious, fatigued and had a skin condition that was worsening.

From the 12 participants completing the intervention, 1 was unable to attend the final interview as they finished participation the week lockdown started in the UK and were living in sheltered accommodation making it impossible for the process evaluation team to make contact.

### Adverse events

No serious adverse events were reported for this study. One participant was admitted to hospital at week 8 of participation for reasons unrelated to the study. Their participation was suspended for 4 weeks and later resumed in agreement between the participant, their family and the researchers.

On two occasions, participants reported feeling unwell and took 1 or 2 days off the intervention. They did not record their activity and no messages were sent to them during this period.

Two instances of data loss occurred, 5 weeks of data were lost for one participant and 1 week for another. Failure to sync data between the local devices (watch and smartphone) and the server accounted for these instances. This did not have an effect on the experience for the participants as they were able to access their data locally on their devices.

### Acceptability and usability of OnTrack
#### Activity tracking

Compliance rate with activity tracking was measured by dividing the total number of days on the intervention by the total number of days the participant recorded their arm activity on the smartwatch. On average, participants were on the intervention for 83 days (min 76, max 84) and recorded their activity on 71 days (min 47, max 84) or the equivalent of 6.1 days per week for a compliance rate of 86% (figure 2).

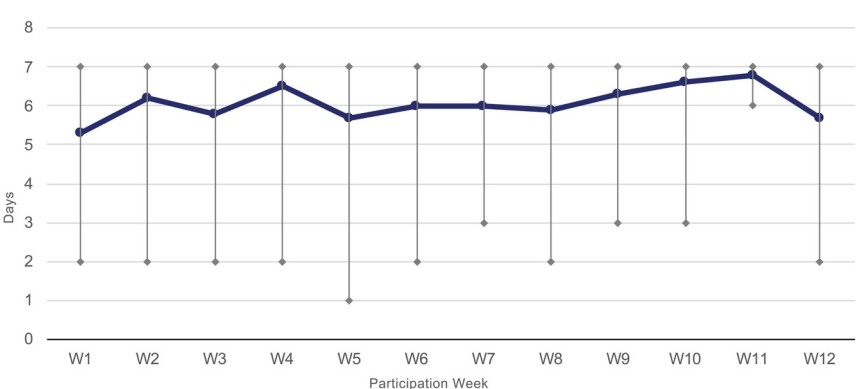

**Figure 2** Average days recording per week (min, max).

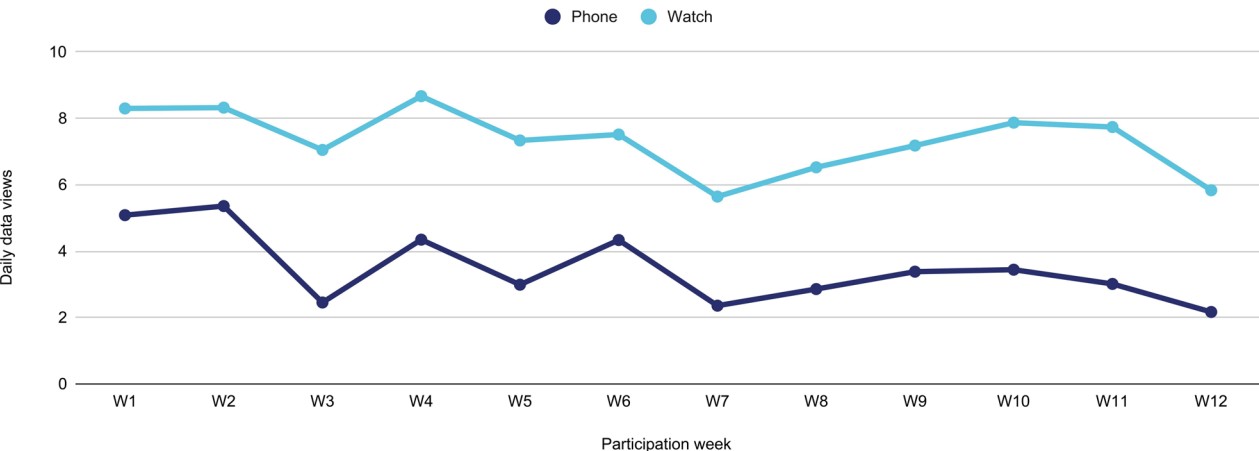

**Figure 3** Average daily data views on phone and watch.

Furthermore an indication of usability and engagement with the intervention could be gathered by measuring the number of daily data interactions participants had using the applications on the smartphone and watch. Figure 3 shows how participants kept track of their activity data on the watch by looking at their activity data a mean of 7.5 times per day (min 4.5, max 12.3) and 3.5 times daily on the smartphone application (min 1, max 9.4).

A 63% mean increase in activity (min −1%, max 864%) could be observed in all participants when comparing the start and end of participation—or the equivalent of 4.3 more hours of activity per week. Participants partially attributed their motivation to do more activity to having a daily target of minutes to achieve. On average, participants increased their target by 5% every session and managed an overall increase of 75% (min 18%, max 200%) between weeks 1 and 12, reaching their target on 3.1 days every week (figure 4).

### Messages and educational content

We measured the way participants engaged with messages, their views were explored during interviews. One participant was deemed cognitively unable to engage with messaging content, this participant was excluded from the messaging component of the intervention.

Overall, an average of 1.5 messages were sent per day and they were opened 1.3 times per day (figure 5). Engagement with messages was lower than expected. One possible explanation is that messages were only sent to the smartphone and no notifications were received on the watch, if the participant did not have the phone with them they would miss the message notification. In addition, most participants felt that carrying the phone was sometimes a burden and so the OnTrack phone was left behind for long portions of the day.

### Correlation between measured activity and application analytics

Correlations between recorded patient activity and application usage analytics were calculated in order to determine if these variables could be of interest for future analysis. There was a significant correlation between patients' activity and their daily activity target ($p < 0.001$, $R = 0.164$), however, this correlation did not appear to be significant when compared with the weekly activity target ($p = 0.323$, $R = -0.035$). There was no significant association with the number of times patients saw their activity graph on their watch ($R = -0.035$, $p = 0.323$). We did not identify a significant difference (Wilcoxon signed rank test,[29] $p = 0.28$) in activity minutes based on whether or not patients accepted their intention for the day (yes=91.7

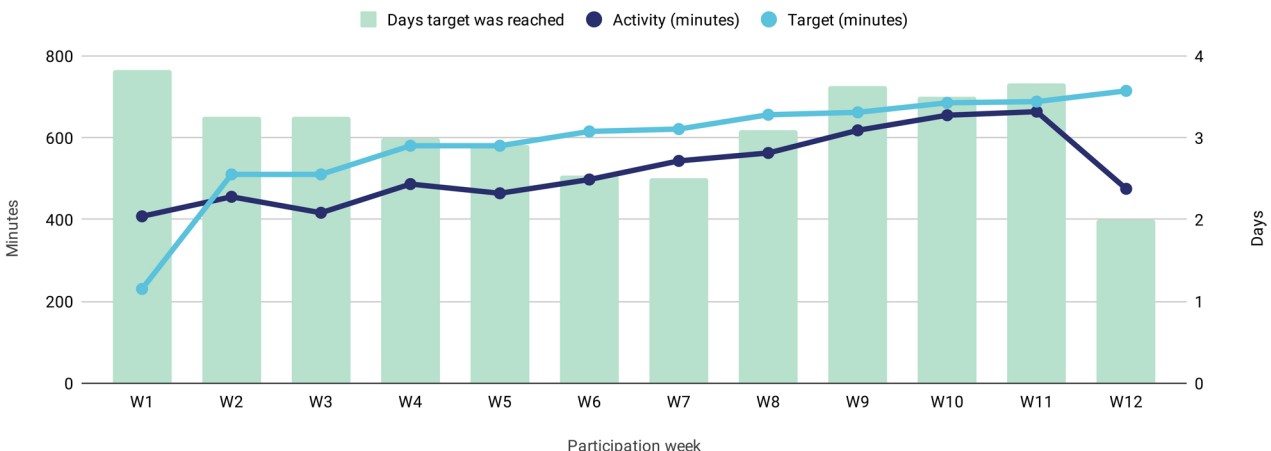

**Figure 4** Weekly activity, target and number of days the target was reached.

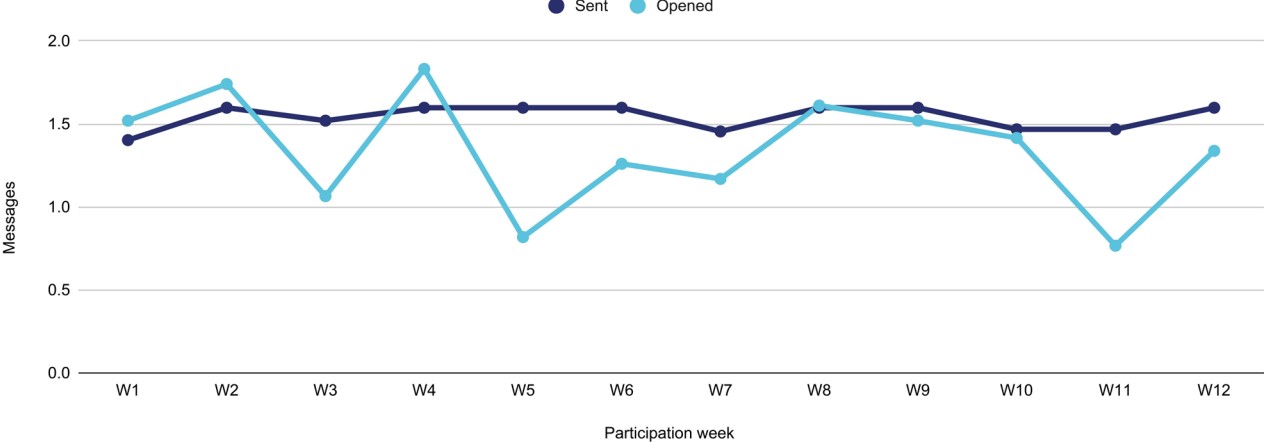

**Figure 5** Overall messages sent/opened (daily average).

min, no answer=64.4 min). Finally, there were positive and significant correlations with the number of times patients viewed messages sent (p<0.001, R=0.201), and the number of links opened (p=0.002, R=0.107).

### Views on the intervention and study procedures
#### Participant views
Participants provided feedback on the study procedures, they felt that recruitment procedures were acceptable and that the information they received was appropriate. Some participants felt that the amount of assessments conducted could be tiring and recommended either reducing them or conducting them over two or more sessions. The activity sensors worn on both wrists during the first and last weeks of participation were uncomfortable for some participants and some required help to put them on.

An in-depth analysis on the experience of participating and using the intervention is reported in a process evaluation publication. In summary, participants felt OnTrack was working as expected and intended. They expressed feeling well supported to use the technology, even without prior experience. They reported feeling that the activity targets were motivating, although they would welcome more specific activity recommendations alongside targets. All participants mentioned that the quality of the coaching was an important part of creating a positive experience. Participants found remote sessions acceptable and reported valuing the contact during lockdown.

#### Therapist views
Therapists contributing to screening and recruitment completed an online survey exploring three themes: the time spent performing study procedures; the intervention and the potential benefits to stroke survivors; and views on patient engagement needed to use OnTrack. 6 of 13 therapists completed the survey (46.2% completion rate).

Overall, therapists felt that the time they spent in study procedures was in line with expectations, and that the tasks performed were acceptable.

On The potential benefits of the intervention, therapists thought that the intervention could be beneficial in general terms and that it could be of most benefit to stroke survivors with motor and sensory impairment. Benefits to patients with arm neglect were less clear but tended to the positive.

Therapists agreed that the intervention could help motivate patients into performing more activities with their impaired arm and support them in the self-management of their recovery. However, some respondents felt that patients who are less engaged with their recovery may not fully benefit from the intervention.

### Clinical outcomes
A wide range of outcome measures were collected (table 4) in order to assess their suitability and feasibility for use during a future definitive study. Pandemic restrictions brought forward new considerations; for example, despite being a widely used assessment for arm impairment after stroke, the FMA-UE was only used during phase 1 of the study as it was impossible to perform once the study moved to a remote format of delivery. On the other hand, the motor activity log (MAL) can be performed remotely and has proven to be a reliable and valid measure of outcome from rehabilitation and of functional status in patients with arm impairment post stroke.[30] An estimate sample size of 46 was calculated using the MAL's reported minimal clinically important difference[31] with 90% power and two-sided alpha of 5%.

A member of the research team was responsible for completing all assessments either in person (phase 1) or remotely (phase 2). Outcomes are reported for the 12 participants completing the intervention.

#### Correlations between measured activity and outcome measures
Several outcome variables were modelled (table 5). Individual models were created for each of the following variables: left-right arm usage ratio improvement, which measures whether or not the patients' difference in arm usage improved by the end of the study (this model also included an additional independent variable which

**Table 4** Outcome measures at baseline and follow-up points, mean and (SD)

| Outcome | Baseline | Halfway* | 14 weeks |
|---|---|---|---|
| PAM | 69.7 (17.8) | 65.8 (14.8) | 68.1 (10.3) |
| FMA-UE† | 37.7 (17.2) | 39.0 (20.1) | 36.4 (22.4) |
| MAL‡ | 2.00 (1.4) \| 2.26 (1.5) | 2.94 (1.2) \| 3.09 (1.2) | 3.24 (1.3) \| 3.17 (1.2) |
| mRS | 2.8 (1.1) | 2.5 (0.8) | 1.9 (0.9) |
| VAS (pain) | 0.8 (1.0) | 2.5 (1.9) | 2.8 (2.3) |
| MoCA§ | 22.6 (6.9) | 24.0 (5.1) | 25.3 (1.0) |
| MoCA¶ | 17.8 (3.4) | 19.0 (1.9) | 18.3 (2.4) |
| AT | 0.2 (0.4) | 0.1 (0.3) | 0.1 (0.3) |
| EQ-5D-5L index | 0.462 (0.3) | 0.585 (0.1) | 0.606 (0.2) |
| EQ-5D-5L VAS | 57.3 (20.7) | 72.7 (9.8) | 74.2 (15.1) |
| SUS | | | 84.6 (13.1) |

* Performed at week 8 for phase 1 participants; week 7 for phase 2.
†Only performed during phase 1. Participants who started in phase 1 but finished their participation after lock down, did not complete this measure subsequently.
‡How much score | How well score
§Full version applied before lockdown measures (scores out of 30).
¶Telephone version applied after lockdown measures (scores out of 22).
AT, Albert's Test; EQ-5D-5L, EuroQol-5 Dimensions-5 Level; FMA-UE, FuglMeyer Upper Extremity; MAL, Motor Activity Log; MoCA, Montreal Cognitive Assessment; mRS, Modified Rankin Scale; PAM, Patient Activation Measure; SUS, System Usability Scale; VAS, Visual Analogue Scale (pain).

captured whether or not the patients' impaired arm was their dominant arm, model fit p<0.001), Patient Activation Measure (model fit p=0.002), VAS (model fit p<0.001), MAL—How much score (model fit p<0.001), MAL—how well score (model fit p<0.001), modified Rankin Scale (model fit p<0.001), Montreal Cognitive Assessment (model fit p=0.158), EQ-5D-5L index (model fit p<0.001), EQ-5D-5L VAS (model fit p<0.001).

### Fidelity of intervention delivery

Fidelity of delivery was assessed through the observation of coaching sessions. Observations during the first half of the study identified that the delivery of the coaching sessions was inconsistent with how participants were experiencing their recovery. This resulted in the intervention team consistently deviating from the coaching plan in response to participant needs.

A workshop involving the PPI group was planned at the study half-way point to refine the delivery of the intervention for phase 2. A summary of findings from the half-way workshop suggested that overall, the OnTrack intervention was working as intended, however the following points were observed:

► Participants reported a lack of clarity about whether the focus of the intervention was to improve hand and arm movement, or to improve activity more generally.

► In coaching sessions, the discussions relating to the activity target were distinct from those relating to individual functional goals, and the self-management strategies were not easy to observe.
► The activity targets were regarded as motivating.
► Participants wanted more specific activities as targets, for example, holding a cup, making toast.
► Participants valued facts and useful information in messages.
► The coaches were seen as central—the language and strategies they use is key to promoting self-management.

In response to these findings, a more fluid and flexible approach to coaching was developed. The new guide included prompts for supporting self-management during all sessions, as opposed to being session specific, it encouraged focusing more on discussing the meaningful activities that participants would like to use their arms for, and using the activity data to support this (a table describing these changes is provided as a online supplemental file).

Delivery of the intervention during phase 2 was affected by the coronavirus pandemic. All sessions were delivered remotely, participants stated that this was acceptable and welcomed the contact with coaches during lockdown.

### DISCUSSION
### Summary of principal findings

This study aimed to assess the acceptability of a novel digital intervention to support arm and hand rehabilitation after stroke and the feasibility of conducting a definitive trial into its effectiveness. The study met its objectives in assessing the acceptability and feasibility of the intervention and trial procedures. Despite being affected by the pandemic, recruitment to the trial was in line with prestudy expectations as described in the protocol's sample size calculation,[23] as well as participant retention rates. Appropriate outcome measures and sample size estimations for an RCT were analysed. Qualitative data were gathered showing that the intervention was usable and acceptable to study participants, but most importantly, it highlighted the components of OnTrack that were perceived as most valuable to them.

### Strengths and weaknesses of the study

Findings from this study have shown a positive way in which wearable technology and self-management coaching could be combined to improve rehabilitation outcomes by helping participants identify opportunities for self-practice and become more engaged in their recovery. Previous studies involving wearable technologies to encourage more practice at home have shown positive results[22 32]; similarly self-management approaches have been shown to obtain better outcomes for stroke patients.[17–20] To our knowledge, this is the first study aiming to combine these two approaches to provide a

 

**Table 5** Outcome measures model estimates

Models were created by using the outcome variable as a dependent variable, and using the variables listed under the 'model variables' as covariates. 'Activity' quantifies segments of 30 min of daily activity, 'time' determines the no of days since each patient started recording their activity, and 'dominant arm' was used to determine whether or not the patient suffered a stroke on their dominant arm.

| Outcome | Model adjusted $R^2$ | Model fit p value | Model variables | Estimates | Estimate p value |
|---|---|---|---|---|---|
| Left-Right ratio improvement* (Ref. category: No improvement) | 0.169† | <0.001 | Activity<br>Time<br>Dominant arm (Ref. category: No) | 1.005‡<br>0.995‡<br>6.684‡ | <0.001<br>0.116<br><0.001 |
| PAM | 0.012 | 0.002 | Activity<br>Time | −0.653<br>−0.022 | 0.002<br>0.290 |
| VAS | 0.135 | <0.001 | Activity<br>Time | −0.056<br>0.29 | 0.025<br>0.000 |
| MAL—how much | 0.452 | <0.001 | Activity<br>Time | 0.304<br>0.012 | <0.001<br><0.001 |
| MAL—how well | 0.433 | <0.001 | Activity<br>Time | 0.324<br>0.001 | <0.001<br><0.001 |
| mRS | 0.267 | <0.001 | Activity<br>Time | −0.161<br>−0.007 | <0.001<br><0.001 |
| MoCA | 0.002 | 0.158 | Activity<br>Time | 0.002<br>0.000 | 0.272<br>0.160 |
| EQ-5D-5L index | <0.001 | 0.161 | Activity<br>Time | 0.020<br>0.002 | <0.001<br><0.001 |
| EQ-5D-5L VAS | 0.226 | <0.001 | Activity<br>Time | 1.252<br>0.241 | <0.001<br><0.001 |

*Left-right ratio improvement was measured using a logistic regression model, as opposed to other variables which were modelled using a linear model.
†Pseudo $R^2$ (McFadden).
‡Estimated for the left-right ratio improvement are show as ORs.
EQ-5D-5L, EuroQol-5 Dimensions-5 Level; MAL, Motor Activity Log; MoCA, Montreal Cognitive Assessment; mRS, modified Rankin Scale; PAM, Patient Activation Measure; VAS, Visual Analogue Scale.

viable intervention that can be delivered in conjunction with usual care.

The coronavirus pandemic significantly affected the running of this study. In the first instance, a suspension in recruitment was necessary, followed by an amendment to the recruitment and consenting procedures. In addition, the adaptation to lockdown restrictions accelerated the need for remote delivery of OnTrack, this, however, had little impact on participants' experiences. As explored in the context of DTx,[22] this may represent an opportunity to challenge conventional clinical practice and may bring a more direct access to services for patients. Furthermore, it may bring additional benefits such as closer monitoring at a lower cost and optimisation of clinician/patient interactions. Future studies should help provide answers to these questions.

The independent process evaluation provided an opportunity to understand the different mechanisms of impact. Allowing for refinement at the half-way point enabled the researchers to incorporate feedback to modify the way OnTrack was delivered. The engagement of the PPI group enabled this refinement to be implemented through a user-centred process. However, the process evaluation also found that the messaging component of the intervention received mixed reviews by participants. On the one hand, the practical information sent was perceived as valuable, but the tone of voice and frequency of other messages was questioned by some participants. This should be revised for a future version of OnTrack.

Due to pragmatic reasons, the study could not cater for longer-term participant follow-up. During interviews, participants expressed they would value a check some time after the programme ended. A future study should consider incorporating longer-term follow-up to understand if participants continue to apply self-management principles.

### Unanswered questions and future research

While randomisation was not possible during this study, participants were asked if they would be willing to participate in a randomised study. Most participants said that randomisation would not influence their decision to participate, however, more consideration is needed to understand the appropriate design of control and intervention arms.

The messaging component of this intervention needs further refinement and co-design in collaboration with

stroke survivors. While, for the most part, the messages are seen as a valuable component, their design requires a better understanding of the content, context, frequency and tone of voice in which they are delivered.

The intervention was delivered over 12 weeks (plus 2 weeks of assessments at each end), however questions remain as to whether OnTrack could be delivered over shorter or longer periods and how this would affect the design of the coaching component. The coaching component itself requires further thinking in how it may be delivered at scale, the training required, and other tools needed to support this role. For this trial, coaching guidelines were used and coaches had access to a digital dashboard showing participants' activity. Future development and studies should focus on understanding these components from the perspective of therapy teams. In addition to this, it will also be necessary to provide answers regarding the economic impact that the intervention may have when delivered as an adjunct to standard care in a NHS such as the UK NHS.

## Conclusion

This study aimed to assess the feasibility, acceptability and safety of the OnTrack intervention when used to support arm and hand rehabilitation in acute and home environments. The results obtained through this study indicated that participants found the intervention to be safe, usable and acceptable. Moreover, the study's feasibility objectives were met and provide a basis to support further investigation to demonstrate the intervention's clinical effectiveness through a RCT.

## Original protocol

https://bmjopen.bmj.com/content/10/3/e034936.

## Author affiliations

[1]Helix Centre, Imperial College, London, UK
[2]Big Data and Analytical Unit, Institute of Global Health Innovation, Imperial College, London, UK
[3]Hyperacute Stroke Unit, Imperial College Healthcare NHS Trust, London, UK
[4]Faculty of Health, Social Care and Education, Kingston University and St George's University, London, UK
[5]Institute of Global Health Innovation, Imperial College, London, UK

Acknowledgements The authors would like to acknowledge the contributions of all the members of the PPI group; all participants involved; Peter Wilding, Meenakshi Nayar and Davina Richardson and therapy teams at ICHT; therapy teams at CLCH; Ruth Nicholson for her support in ethics applications and procedures; Steve McAteer and Eleni Daniels for their guidance on the BRC program; Dr Hendramoorthy Maheswaran for his help reviewing the manuscript.

Contributors AD was principal investigator and grant holder and oversaw the project along with DD. GF, EG and LH developed the intervention and conceived of the study. GF, EG and FJ initiated the study design and ET and ML helped with further refinement. AL-J led on PPI. EG and GF were responsible for delivering the intervention and data collection. JC had participant screening oversight. FJ and ET were responsible for the process evaluation. ML and RFC conducted the primary statistical analysis. GF is responsible for the overall content as the guarantor. All authors contributed to this manuscript and approved the final version.

Funding This study was funded by the NIHR Imperial Biomedical Research Centre (BRC), grant 1215-20013.

Disclaimer The views expressed are those of the author(s) and not necessarily those of the NIHR or the Department of Health and Social Care.

Competing interests None declared.

Patient and public involvement Patients and/or the public were involved in the design, or conduct, or reporting, or dissemination plans of this research. Refer to the Methods section for further details.

Patient consent for publication Not applicable.

Ethics approval This study involves human participants and was approved by the NHS Health Research Authority, Health and Care Research Wales, and the London—Surrey Research Ethics Committee (ref. 19/LO/0881).

Provenance and peer review Not commissioned; externally peer reviewed.

Data availability statement Data are available on reasonable request. All data relevant to the study are included in the article or uploaded as online supplemental information. All data relevant to the study are included in the article or uploaded as online supplemental information. Additional data relevant to the study are available on reasonable request and should be directed to the corresponding author.

ORCID iDs
Gianpaolo Fusari http://orcid.org/0000-0002-7263-3398
Roberto Fernandez Crespo http://orcid.org/0000-0002-2328-3782
Elizabeth Taylor http://orcid.org/0000-0002-4596-823X

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
