## [Reviewer comments · BMJ Open]

ARTICLE DETAILS

TITLE (PROVISIONAL)	What is the feasibility and patient acceptability of a digital system for arm and hand rehabilitation after stroke? A mixed methods, single-arm feasibility study of the 'OnTrack' intervention for hospital and home use.
AUTHORS	Fusari, Gianpaolo; Gibbs, Ella; Hoskin, Lily; Lawrence-Jones, Anna; Dickens, Daniel; Fernandez Crespo, Roberto; Leis, Melanie; Crow, Jennifer; Taylor, Elizabeth; Jones, Fiona; Darzi, Ara

VERSION 1 – REVIEW

REVIEWER	Luigi Lavorgna I Clinic of Neurology University of Campania Luigi Vanvitelli
REVIEW RETURNED	26-Mar-2022

GENERAL COMMENTS	The study by Fusari et al. explored feasibility and acceptability of OnTrack, a digital intervention supporting arm and hand rehabilitation in acute and community settings. The manuscript is highly topica. Objectives and methods are clearly presented. Some minor concerns: 1. On Track system belong to the chapter of Digital Therapeutics (DTx). In the introduction section, the authors should highlight this. Furthermore, in the discussion section, they should discuss how digital therapeutics have emerged in the last year and how they could impact the future of stroke rehabilitation (this is widely reviewed here: PMID: 34018047).2. A relevant issue concerning the implementation of DTx in clinical practice is the economic impact on the healthcare system. The authors may mention this aspect in relation to the evaluated system.
---

REVIEWER	Federica Bressi Campus Bio-Medico University Hospital
REVIEW RETURNED	12-Apr-2022

GENERAL COMMENTS	The idea and the project are interesting, however there are some parts that need to be reviewed. The fact that you considered all patients instead of divided them into two groups of analysis (pre and post covid- community versus home) could be a bias. Therefore, it's important to better define patients' inclusion and exclusion criteria and study protocol. Therefore reported in the text references to figures or supplementary materials. 2)The abstract should be more technically detailed in all its parts. It is too succinct. Therefore, it's important to better specify the inclusion criteria and report the main results.
---

	4) It is necessary to better specify the identikit of the patient to whom the study is addressed. For example, indicating for each outcome the cut-off within which patients are inserted. In addition, it is useful to detail how the device is used. For example, is everyone asked to use it in the same way and at the same time every day or are they free to use it as they wish? I also don't understand if the treatment is composed of 12 or 14 weeks. Finally, specify in the text the reference to supplementary material to identify what self-management coaching consists of. 6) The outcomes used are well defined. Have you also considered mood scales such as depression and anxiety scales? 7) It is necessary to indicate which statistical tests were used and also whether intention to treat was assessed. Second, it is important to specify how many patients have followed all the way from home and how many in the community. Furthermore, specify if there are differences between the two groups at the baseline and if they all underwent rehabilitation treatment in these 14 weeks. 9-10) Explain better on how many patients the evaluations were actually assessed, to facilitate the reader in understanding the data. To better understand those who have completed the entire study compared to those who have made a partial path. 11) The discussion is very poor, it should be expanded on the basis of the literature, the results obtained and the considerations made in the previous points. Furthermore, it is necessary to explain what is meant by "pre-study expectation", as they are not reflected in the text. Therefore, the conclusions are reported only in the abstract and they are too strong for results obtained on 11 patients. You could say that a RCT design could confirm these preliminary results. 12) Why are some of the limitations and strengths reported after the abstract? Furthermore, the diversity of the setting in which the project was carried out could have influenced the results obtained, should be included among the limitations.
--	---

VERSION 1 – AUTHOR RESPONSE

Reviewer	Comment	Response
Dr. Luigi Lavorgna, University of Campania Luigi Vanvitelli	On Track system belong to the chapter of Digital Therapeutics (DTx). In the introduction section, the authors should highlight this. Furthermore, in the discussion section, they should discuss how digital therapeutics have emerged in the last year and how they could impact the future of stroke rehabilitation (this is widely reviewed here: PMID: 34018047).	We thank the reviewer for their comments and agree with their view. We have made a connection between our intervention and the DTx field in the introduction (p3 ln31-33); we have also added a paragraph on this matter in the Discussion section (p14 ln25-29).
	A relevant issue concerning the implementation of DTx in clinical practice is the economic impact on the healthcare system. The authors may mention this aspect in relation to the evaluated system.	We thank the reviewer and acknowledge that the economic impact of a DTx intervention is of utmost importance for decision making towards implementation. Although an economic evaluation was not in scope for the present study, our planned future work includes a cost-consequence analysis to

		be carried out in 2023 in the UK NHS. We have added a reference to this in the Discussion section (p15 ln16-18).
Dr. Federica Bressi, Campus Bio-Medico University Hospital	The fact that you considered all patients instead of divided them into two groups of analysis (pre and post covid- community versus home) could be a bias. Therefore, it's important to better define patients' inclusion and exclusion criteria and study protocol. Therefore reported in the text references to figures or supplementary materials.	We thank the reviewer for their comments and take onboard their feedback. We have made a clearer distinction of what we mean by “community settings” by changing this to “home setting” throughout the manuscript. We have also made a clearer reference to the inclusion / exclusion criteria as detailed in the study protocol (p4 ln15-18). Regarding the risk of bias from considering all patients as one group, the main purpose of the study was to assess the feasibility of recruitment and understand retention & withdrawal rates; all patients were recruited in the acute setting and were subsequently followed up at their own home when discharged. The main difference between pre/post covid participation was that instead of starting the intervention in the acute setting, post-covid participants started directly at home. We therefore consider that all clinical outcome measure results are only indicative and were performed to assess their suitability for being used in a RCT.
	The abstract should be more technically detailed in all its parts. It is too succinct. Therefore, it's important to better specify the inclusion criteria and report the main results.	We thank the reviewer for their feedback. We have added to the abstract and provided more detail on the participants (p2 ln4-5); added a mention of participants recruited pre and post Covid (p2 ln15-17) in the main results; and have also included a note on the suitability of outcome measures for a RCT (p2 ln20) in the main results section. We have also ensured that these changes adhere to BMJ guidance for

		abstracts.
	It is necessary to better specify the identikit of the patient to whom the study is addressed. For example, indicating for each outcome the cut-off within which patients are inserted. In addition, it is useful to detail how the device is used. For example, is everyone asked to use it in the same way and at the same time every day or are they free to use it as they wish? I also don't understand if the treatment is composed of 12 or 14 weeks. Finally, specify in the text the reference to supplementary material to identify what self-management coaching consists of.	The authors thank the reviewer for their comments and questions. A definition of eligible participants has been added to the Methods/Participants section (p4 ln14-18). We have expanded on details about how and when the smartwatch was used by participants (p5 ln1-2). We have clarified what the participation entailed for the 14 weeks (p4 ln30-33). Table 2 also provides a detailed description of outcome measures used. A reference to the self-management coaching components is now provided in the Intervention section (p5 ln9).
	The outcomes used are well defined. Have you also considered mood scales such as depression and anxiety scales?	We thank the reviewer for their positive feedback and suggestions to improve our research. Whilst we have not directly measured anxiety and depression during this study, one of the outcome measures used, the EQ-5D-5L, captures quality of life related domains, including anxiety and depression. We appreciate that there could be value in using a specific measure for anxiety and depression, such as the Hospital Anxiety and Depression Scale (HADS), and we will consider this in our future research.
	It is necessary to indicate which statistical tests were used and also whether intention to treat was assessed. Second, it is important to specify how many patients have followed all the way from home and how many in the community. Furthermore, specify if there are differences between the two groups at the baseline and if they all underwent rehabilitation treatment in these 14 weeks.	We thank the reviewer for their observations. We have added a paragraph to explain the statistical methods used in the Methods section under Analysis (p7 ln25-28). As previously mentioned, all participants were recruited in the acute setting and followed up at home to complete their

		participation. Details regarding numbers of participants recruited before and after Covid disruptions are provided in the Results section under Recruitment (p8 ln6-12). Differences between these groups can not be established as outcome measure results are indicative and only performed to assess their suitability in a RCT. All participants underwent some form of rehabilitation treatment by a UK NHS acute and/or community provider, ranging from in-person ESD to remote community neurorehabilitation. However, the study did not collect detailed information on the type of rehabilitation that was provided. We have highlighted this in the manuscript (p8 ln23-25)
	Explain better on how many patients the evaluations were actually assessed, to facilitate the reader in understanding the data. To better understand those who have completed the entire study compared to those who have made a partial path.	We thank the reviewer for this comment. The study reports on the sample of patients who completed the intervention (n=12), no results are provided for participants who withdrew from the study. We have attempted to make this clear in the manuscript (p12 ln2).
	The discussion is very poor, it should be expanded on the basis of the literature, the results obtained and the considerations made in the previous points. Furthermore, it is necessary to explain what is meant by "pre-study expectation", as they are not reflected in the text. Therefore, the conclusion are reported only in the abstract and they are too strong for results obtained on 11 patients. You could say that a RCT design could confirm these preliminary results.	We thank the reviewer and we take their feedback on board. We have expanded the discussion to include how our results compare to the literature (p14 ln17-21). We have also added a paragraph to highlight the relevance of this intervention in relation to the field of digital therapeutics, as suggested by the other reviewer (p14 ln25-29). A line has been added to refer the reader to the study protocol outlining the pre-study recruitment expectations (p14 ln9-10). A conclusion section has been added in the Discussion (p15

		In19-24)
	Why are some of the limitations and strengths reported after the abstract? Furthermore, the diversity of the setting in which the project was carried out could have influenced the results obtained, should be included among the limitations.	We thank the reviewer for their questions and feedback. Our manuscript follows the editor's guidance which requests up to 5 short statements on strengths and limitations of the study. Regarding setting diversity, the study was planned to be delivered in these varied settings. The process evaluation conducted in parallel to the study found that there was good consistency in the delivery of the intervention across participants and settings, therefore we don't perceive this as a limitation but accept that results may be influenced by this.